# Furosemide and Ductus Arteriosus Closure in Very-Low-Birth-Weight Preterm Infants: A Comprehensive Retrospective Study

**DOI:** 10.3390/children11050610

**Published:** 2024-05-20

**Authors:** Chi-Mei Kuo, Pin-Chun Su, Shu-Ting Yang, Hao-Wei Chung, Hsiu-Lin Chen

**Affiliations:** 1Department of Pediatrics, Kaohsiung Medical University Hospital, Kaohsiung City 80756, Taiwan; 1080232@kmuh.org.tw (C.-M.K.); 1020407@gap.kmu.edu.tw (P.-C.S.); 1000410@kmuh.org.tw (S.-T.Y.); 1000461@gap.kmu.edu.tw (H.-W.C.); 2Department of Respiratory Therapy, College of Medicine, Kaohsiung Medical University, Kaohsiung City 807378, Taiwan

**Keywords:** furosemide, ductus arteriosus, spontaneous closure, very-low-birth-weight, preterm, infants, retrospective, patent ductus arteriosus

## Abstract

Ductus arteriosus closure may be delayed in preterm infants, and prostaglandin, a vasodilator, can affect ductal patency. Furosemide can increase renal prostaglandin synthesis, so its net effect on patent ductus arteriosus (PDA) is uncertain. Our goal is to explore the relationship between furosemide and spontaneous ductal closure in very-low-birth-weight preterm infants. Our treatment for PDA involves fluid restriction initially and furosemide administration for hemodynamically significant PDA until closure is confirmed by the echocardiogram. We enrolled 105 infants from 1 January 2019 to 30 June 2022 and evaluated the impact of furosemide on ductal closure, including exposure duration and cumulative dose. There is no correlation between furosemide exposure and spontaneous ductal closure (*p* = 0.384). Furosemide exposure does not delay the postmenstrual age at which spontaneous ductal closure occurs (*p* = 0.558). The time for spontaneous ductal closure is positively associated with furosemide prescription days (coefficient value = 0.547, *p* = 0.026) and negatively with gestational age (coefficient value = −0.384, *p* = 0.062). The prescription of furosemide does not impact the probability or time duration of ductus arteriosus spontaneous closure. The cumulative dose of furosemide has minimal impact on ductal closure. The correlation between furosemide exposure duration and ductal patency duration is likely due to our treatment protocol, with gestational age being a significant factor.

## 1. Introduction

In fetal circulation, the ductus arteriosus (DA), connecting the main pulmonary artery and aorta, allows oxygenated blood to bypass the lungs, and it typically closes shortly after birth [1]. Ductal closure occurs within 72 h after birth in nearly all term infants, but at the same postnatal age, the DA remains patent in approximately 65% of preterm infants with a gestational age (GA) of less than 30 weeks [2]. Among very-low-birth-weight (VLBW) infants (birth body weight (BBW) less than 1500 g), the incidence of patent ductus arteriosus (PDA) is around 30% [3]. Decreased pulmonary resistance leads to left-to-right PDA shunting, causing ventricular overload and pulmonary edema. Clinical signs of a hemodynamically significant PDA (hsPDA) include an increased need for respiratory support, widened pulse pressure, cardiomegaly and/or pulmonary edema evident on the chest radiographs, hypotension, and oliguria, and it can cause several adverse consequences, including pulmonary hemorrhage, bronchopulmonary dysplasia (BPD), pulmonary hypertension, necrotizing enterocolitis (NEC), intraventricular hemorrhage, and acute kidney injury [4,5,6,7,8,9].

The management of PDA remains controversial because limited evidence is available [10]. Four main strategies are utilized: fluid restriction and diuretics usage as clinically appropriate, medical intervention involving cyclooxygenase or peroxidase inhibitors, surgical ligation, and transcatheter PDA closure [11,12]. About 60–70% of preterm infants with a GA of less than 28 weeks were treated with medical or surgical therapies [13]. There are a variety of pharmacological interventions for PDA, including cyclooxygenase inhibitors, like indomethacin and ibuprofen, as well as peroxidase inhibitors, such as acetaminophen [14]. About 70 randomized controlled trials (RCTs) were performed to evaluate the pharmacological interventions of a PDA; however, there are no RCTs concluding with definitive options [14]. Although several interventions can be chosen, there has been a trend toward watchful waiting without intervention because of the high likelihood of spontaneous closure [15]. In 2017, Semberova et al. reported that 85% of infants diagnosed with PDA but left untreated experienced spontaneous ductal closure before discharge [16]. One study showed that, compared to active PDA treatment, conservative treatment did not increase the mortality and morbidity in infants born at GA 28–32 weeks [17]. During the waiting period, furosemide is commonly used to reduce pulmonary edema caused by PDA shunting. Furosemide is commonly administered to as many as 50% of hospitalized preterm infants [18], and one study reported that over 24% of infants with hsPDA were exposed to furosemide at some point [19].

In a dose-dependent manner, furosemide can increase renal prostaglandin E2 (PGE2) synthesis in Henle’s loop of rats [20]. The PGE2, a vasodilator, can affect ductal patency in animal models (in lambs and rats) [21,22]. During the perinatal transition, several complex mechanisms make the DA closure, including increased neonatal blood oxygenation, pressure changes from placental removal, and decreased serum prostaglandin levels [11]. There is a concern that the renal-originated furosemide exposure-induced PGE2 may hinder the ductal closure. The US Food and Drug Administration notes that furosemide may promote ductal patency [23]. The net effect of furosemide on hsPDA is unpredictable; it decreases pulmonary edema while simultaneously potentially inhibiting DA closure [14]. The guideline of the North West Neonatal Operational Delivery Network in the UK recommends chlorothiazide over furosemide for PDA management due to risks of delayed closure, nephrotoxicity, and ototoxicity [24]. However, among newborns, furosemide remains the most frequently prescribed and studied diuretic. Loop diuretics, compared to other agents, are more effective due to their rapid onset and higher diuresis [25].

Several studies have assessed furosemide’s overall impact on hsPDA physiology, yet conflicting evidence exists. The studies of Green et al. in 1980 showed higher PDA rates in infants treated with furosemide [26,27,28]; however, in an RCT of infants with respiratory distress syndrome, furosemide administration improved respiratory condition without increasing PDA rates [29]. As time progresses, the trend in managing PDA is gradually shifting to a more conservative approach, which potentially increases furosemide usage. Our aim is to explore the relationship between spontaneous ductal closure and furosemide exposure in VLBW preterm infants without medical or surgical intervention for hsPDA.

## 2. Materials and Methods

We collected data on infants with a BBW of less than 1500 g and a GA of less than 32 weeks who were admitted to the neonatal intensive care unit (NICU) of Kaohsiung Medical University Hospital from 1 January 2019 to 30 June 2022 (Figure 1). The exclusion criteria included infants with a BBW of 1500 g or more, those with a GA of 32 weeks or more, those who survived for less than 24 h after birth, those having incomplete medical records, those diagnosed with cyanotic congenital heart disease, those receiving prostaglandin therapy after birth, those with chromosomal anomalies, and those with acquired heart disease such as infective endocarditis and twin-twin transfusion syndrome. The institutional review board of Kaohsiung Medical University Hospital approved the use of the data (approved number: KMUHIRB-SV(I)-20230077, date of approval: 8 September 2023). Given the retrospective nature of the study, informed consent was not required.

In our NICU, the protocol for performing echocardiograms on preterm infants involves conducting a general survey echocardiogram for all preterm newborns mostly between 3 and 7 days after birth. Exceptions are made for those with congenital cyanotic heart disease, persistent pulmonary hypertension of the newborn, cardiac tamponade, arrhythmia, or other critical conditions that necessitate an immediate echocardiogram to assess heart function. Following initial assessments, and under the provisions of Taiwan’s National Health Insurance, we schedule monthly follow-up echocardiograms until the ductal closure is confirmed. Ductal closure is defined by the absence of identifiable flow in the DA using color Doppler ultrasound. If there is a significant deterioration in an infant’s clinical condition that requires a reevaluation of heart function, an additional echocardiogram is performed. All pediatric cardiologists conducting these echocardiograms are well-trained and experienced in the technique. Since 2011, our policy for treating PDA has tended toward conservative treatment. We initially restrict fluid intake (between 140 and 150 mL/kg/day), and then if an infant has symptoms and signs suggestive of hsPDA, furosemide is administered regularly until the DA is closed. We define hsPDA as PDA accompanied by symptoms/signs of heart failure, including increased oxygen demand, respiratory acidosis, pulmonary edema, cardiomegaly, oliguria, pitting edema, hypotension, and inadequate body weight gain. We rely on clinical symptoms and signs rather than echocardiographic parameters to define hemodynamic significance. Consequently, we do not have any echocardiographic parameters that the team uses to determine hemodynamic significance. If the infants exhibit signs and symptoms of heart failure or worsening respiratory status after fluid restriction, furosemide is administered primarily to improve respiratory status or in response to heart failure symptoms. The initial dose of furosemide administered is 0.5 mg/kg/dose every 12 to 8 h, depending on the attending neonatologists’ judgement. We adjust the dose, either increasing or decreasing it, based on the patient’s clinical condition, including improvements in oliguria, pitting edema, pulmonary edema, and body weight gain, among other factors. In our NICU, furosemide is also prescribed following blood transfusion or albumin supplement. This study utilized two definitions of furosemide exposure: (I) in infants with PDA, any exposure to intravenous or oral furosemide after birth until the day of the first echocardiographic confirmation of spontaneous ductal closure, and (II) in those without PDA, any exposure to intravenous or oral furosemide after birth until the day of discharge. We do not use classical prostaglandin inhibitors to manage PDA in our unit. Consequently, in this study, infants without furosemide exposure did not receive prostaglandin inhibitors after birth, taking into account the potential side effects associated with such treatments.

We collected demographic data for enrolled survived infants, including sex, BBW, GA, 1st and 5th minute Apgar score, exposure to antenatal steroids, classification as small for gestational age or large for gestational age, and whether they required intubation at birth. We used the revised Fenton growth chart for preterm infants to identify their birth percentile [30]. Additionally, we recorded the timing (postnatal day and postmenstrual age (PMA) of the first echocardiography-documented spontaneous closure of PDA. For infants diagnosed with PDA, the cumulative doses and duration of furosemide exposure were calculated. The collected data on clinical outcomes included intraventricular hemorrhage, BPD, NEC, retinopathy of prematurity (ROP), pulmonary air leak syndrome, periventricular leukomalacia, and post-hemorrhagic hydrocephalus. We adhered to the diagnostic criteria of BPD proposed by Jobe AH et al. in 2001 [31], and the severity of NEC was evaluated using the modified Bell staging criteria [32]. The mortality rates of two groups (enrolled infants with and without PDA) were also calculated, although expired cases were excluded from the demographic data and clinical outcomes.

We used the IBM SPSS statistics version 20 software to execute statistical analysis, and the Chi-squared test, *T*-test, and univariate and multivariate regression analyses were properly performed to analyze data. We used the Chi-squared test and *T*-test to evaluate the differences in characteristics of infants with and without PDA. For those who had PDA with and without furosemide exposure, we evaluated the differences in characteristics by using the Chi-squared test and *T*-test. To analyze the timing of the first documented spontaneous closure of PDA by postnatal day and PMA, we used the *T*-test to compare two groups (infants with PDA with and without furosemide exposure). Additionally, we conducted both univariate and multivariate regression analyses in infants exposed to furosemide to identify variables affecting the time of the first documented spontaneous ductal closure by postnatal day and PMA. The variables tested included furosemide cumulative dose, furosemide prescription days, BBW, GA, 1st and 5th minute Apgar score, sex, antenatal steroid exposure, and having intubation at birth. Regarding the analysis of clinical outcomes mentioned previously, we performed the *T*-tests to identify significant outcomes during hospitalization. Among the significant outcomes, we further utilized both univariate and multivariate regression analyses to identify factors significantly associated with them.

## 3. Results

### 3.1. Mortality Rate of Enrolled Infants

Our retrospective analysis included 105 VLBW infants born at less than 32 weeks of gestation. Of these, nine infants died during their hospital stay. All nine expired infants had PDA, representing 13.0% (9/69) of the cases with PDA. No mortality occurred among infants. The primary causes of mortality of these nine infants were mainly attributed to septic shock (3/9), bilateral tension pneumothorax (2/9), pulmonary hemorrhage (2/9), BPD-related pulmonary hypertension (1/9), and NEC stage IIIA (1/9). Notably, complications associated with PDA contributed to approximately 44.4% (4/9) of the mortality cases.

### 3.2. Characteristics of VLBW Preterm Infants with/without PDA

In 96 surviving infants, 62.5% (60 infants) had PDA. Infants with PDA were significantly more premature (GA: 27.1 weeks vs. 28.4 weeks, *p* = 0.003), and they also had a lower BBW (992.4 g vs. 1174.5 g, *p* = 0.001), a lower 1st minute Apgar score (4.0 vs. 4.8, *p* = 0.025), and a higher intubation rate at birth (56.7% vs. 27.8%, *p* = 0.006) (Table 1).

### 3.3. Characteristics of VLBW Preterm Infants with PDA with/without Furosemide Exposure

Among the 60 surviving infants with PDA, 85.0% (51 infants) were treated with furosemide. Those treated with furosemide were significantly more premature (GA: 26.6 weeks vs. 29.6 weeks, *p* < 0.001), and had a lower BBW (931.7 g vs. 1336.4 g, *p* < 0.001) as well as a lower 5th minute Apgar score (5.7 vs. 7.2, *p* = 0.002). Among 47 VLBW preterm infants with spontaneous ductal closure and exposure to furosemide, the mean cumulative dose was 67.5 mg, and the median cumulative dose was 34.2 mg. The maximum cumulative dose was 323.4 mg, while the minimum was 0.7 mg. There was no significant relationship between furosemide exposure and spontaneous closure of PDA (*p* = 0.384). Infants exposed to furosemide also experienced a longer duration before PDA closure (55.1 days vs. 28.0 days, *p* = 0.022), although there was no significant difference in the PMA at the time of spontaneous ductal closure between the two groups (35.0 weeks vs. 34.1 weeks, *p* = 0.558) (Table 2).

### 3.4. Factors Associated with the Time Taken for Spontaneous Ductal Closure in VLBW Preterm Infants with Furosemide Exposure

Factors influencing the time taken by spontaneous PDA closure in VLBW preterm infants exposed to furosemide were examined. Through univariate regression analysis, we identified factors affecting the postnatal day of spontaneous PDA closure, where four factors (furosemide cumulative dose, duration of furosemide treatment, BBW, and GA) showed significant impact. Further assessment using multivariate regression analysis revealed that the duration of furosemide treatment was positively associated with the time taken for spontaneous ductal closure (*p* = 0.026). However, furosemide cumulative dose, GA, and BBW were not significantly influential (*p* = 0.668, 0.062, and 0.317, respectively) (Table 3).

### 3.5. Clinical Outcomes of VLBW Preterm Infants

As for clinical outcomes, there was no significant difference between infants with and without PDA. However, infants with PDA who were exposed to furosemide exhibited a higher risk of developing BPD and ROP compared to those with PDA but not exposed to furosemide (BPD: *p* = 0.034; ROP: *p* = 0.009) (Table 4).

### 3.6. Factors Associated with BPD and ROP in VLBW Preterm Infants

Further analysis through both univariate and multivariate regression was conducted to identify risk factors for BPD and ROP, as well as to assess the impact of furosemide exposure on these conditions. Univariate regression analysis identified five significant factors associated with BPD (BBW, GA, 1st-minute Apgar score, 5th-minute Apgar score, and furosemide exposure), while for ROP, three significant factors were identified (BBW, GA, and furosemide exposure). Subsequent multivariate regression analysis, however, revealed that only GA was negatively associated with the risk of BPD and ROP (*p* = 0.012 for BPD and *p* = 0.001 for ROP, respectively) (Table 5 and Table 6).

## 4. Discussion

Our retrospective study found no evidence of a correlation between the cumulative dose of furosemide and spontaneous ductal closure, indicating that the administration of furosemide did not influence the natural process of ductal closure. Additionally, our findings reveal that furosemide exposure does not delay the PMA at which spontaneous ductal closure occurs, further supporting the conclusion that it follows its inherent timeline. We observed that the duration of furosemide exposure positively correlated with a longer time to spontaneous ductal closure. This correlation likely reflects our treatment protocol for hsPDA, wherein we continue furosemide administration until the closure of the DA is first confirmed via echocardiography. Concerning echocardiography, the initial echocardiogram for all preterm newborns is conducted between 3 and 7 days after birth in our NICU. Subsequently, in accordance with Taiwan’s National Health Insurance provisions, we perform follow-up cardiac ultrasound monthly until the closure of the DA is confirmed. Given this protocol, the observed statistical association is expected and does not suggest that furosemide directly prolongs the time required for DA closure.

According to our multivariate regression analysis (Table 3), GA appears to be the most likely factor influencing the timing of spontaneous ductal closure, displaying a negative association that is statistically insignificant. We hypothesize that the lack of strong statistical power could be attributed to the limited sample size of our study. Semberova et al. conducted a retrospective study of 280 preterm infants who did not receive any intervention to close PDA, finding that the median time for ductal closure varied by GA: less than 26 weeks (71 days), 26–27 weeks (13 days), 28–29 weeks (8 days), and 30 weeks or more (6 days) [15]. This gradient suggests a negative correlation with GA, aligning with our study’s findings. This finding could be explained by several mechanisms affecting ductal closure in preterm infants. Primarily, the DA in preterm infants is more sensitive to PGE2 compared to term infants, and the metabolic breakdown of PGE2 occurs at a slower rate in earlier stages of gestation [33,34]. Secondly, the response to oxygen, which is mediated through potassium channels [35], plays a crucial role in ductal closure. The reduced expression of these channels in preterm infants leads to a failure of the DA closure under normoxic conditions [36]. Moreover, the immature functionality of the mitochondrial reactive oxygen species system in preterm infants may prevent them from maintaining the contractile state necessary for ductal closure [37]. This immaturity in the reactive oxygen species system suggests preterm infants may lack physiological mechanisms to effectively close the DA in response to postnatal oxygen levels. Finally, the immaturity of smooth muscle myosin isoforms in preterm infants results in a reduced contractile capacity [37]. This diminished ability contributes to the challenge of achieving closure of the DA. Given this reduced contractility, it is so logical and consistent with observed outcomes that the more premature an infant is, the longer time a DA takes to close spontaneously. This observation underscores the inherent physiological limitations faced by preterm infants in achieving ductal closure, further highlighting the influence of developmental maturity on this critical postnatal adaptation. We also speculate that the reason why the total duration of furosemide treatment was found to be a significant predictor for PDA closure in multivariate regression, rather than the cumulative dose of furosemide, can be explained as follows: Since furosemide administration is discontinued once the DA closes, the duration of furosemide treatment may be associated with the time taken for ductal closure. According to our previous study published in 2015 [38], there is a direct relationship between PMA and spontaneous PDA closure in very preterm infants. This suggests that GA is a critical factor for the closure of the DA, rather than the administration or cumulative dose of furosemide, as previously discussed.

In our study, 93.3% (56/60) of the surviving infants diagnosed with PDA achieved spontaneous closure, indicating a high likelihood of natural PDA closure without any medical intervention in this group. Moreover, among infants treated with furosemide, the rate of spontaneous ductal closure was 92.2%, closely aligning with the 100% closure rate observed in infants not exposed to furosemide. This comparison revealed no statistically significant difference between the two groups, suggesting furosemide exposure does not adversely impact the natural course of spontaneous ductal closure. According to the prior research carried out in our NICU by Chen HL et al. in 2015, their study found 96.4% of very preterm infants (those with a BBW of less than 1500 g and a GA of less than 32 weeks) diagnosed with PDA exhibited echocardiographic evidence of spontaneous ductal closure before discharge from the hospital [38]. Semberova et al., in 2017, reported that 85% of VLBW infants with untreated PDA experienced spontaneous closure before discharge [15]. In 2022, Nielsen et al. revealed that among infants with a GA of less than 32 weeks, the rate of spontaneous ductal closure rate within the first year of life was 66%, which increased to 80% by five years post-birth [39]. The rate of ductal closure in our study exceeds those reported in the above-mentioned studies, with the exception of the findings by Chen HL et al. in 2015 [38], which reported a higher spontaneous closure rate during hospitalization. This disparity in closure rates could be attributed to the exclusion of infants who did not achieve ductal closure prior to their demise, representing 13.0% (9 out of 69) of our study population. Our neonatology team supports a conservative approach to managing PDA, as articulated in the study conducted by Chen HL et al. in 2015 [38]. This methodology reflects our commitment to minimizing interventions, favoring natural closure processes when feasible, and likely contributes to our observed outcomes. Furthermore, our assessment of PDA spontaneous closure extended beyond the hospital stay, encompassing all survived infants who achieved closure, whether during hospitalization or through outpatient follow-up. Therefore, our findings represent the natural course of DA closure without any medical or surgical interventions in VLBW preterm infants. This approach underscores the significant potential for spontaneous closure in this demographic, reinforcing the effectiveness of a conservative management strategy.

Based on our study, the PMA at which spontaneous ductal closure occurred revealed the mean PMA for infants treated with the furosemide group was 35.0 weeks, closely aligning with the 34.1 weeks observed in infants not exposed to furosemide. This finding is consistent with a 2015 study conducted in the same unit, which demonstrated in very preterm infants, the cumulative probability of spontaneous ductal closure reached 70% by a PMA of 35 weeks and exceeded 90% by a PMA of 38 weeks [38]. This similarity in PMAs between groups, regardless of furosemide exposure, and the high probability of closure by certain PMAs, highlights the natural progression toward spontaneous closure in this population. A previous study revealed the majority of PDA in preterm infants closed spontaneously by a PMA of 44 weeks when left untreated [40]. Additionally, two other studies explored the progression of infants with PDA after discharge. Herrman et al. observed a median PMA of 48 weeks at which spontaneous closure occurred [41], while Weber et al. reported a mean time for spontaneous closure was 9 months post birth, which was approximately at a PMA of 49 weeks [42]. The PMA at which spontaneous ductal closure occurs appears to vary widely, indicating a discrepancy that may stem from the significantly different frequency of follow-up intervals for PDA across various countries and hospitals. Specifically, within the framework of Taiwan’s National Health Insurance coverage, neonatologists and pediatric cardiologists in our hospital face constraints in scheduling frequent echocardiogram follow-ups for PDA monitoring due to cost considerations. Thus, our standard practice typically involves monthly follow-ups for infants diagnosed with PDA, barring instances of significant clinical deterioration. This approach reflects a balance between clinical necessity and financial constraints, but it potentially affects the observed timing of spontaneous ductal closure in our patient population.

Furosemide is widely used among preterm infants, as evidenced by a retrospective cohort study that included VLBW infants with a GA of less than 32 weeks across 333 NICUs. This study showed about 37% of these infants were exposed to at least one diuretic, with furosemide being the most frequently administered diuretic and 93% of those exposed to diuretics received at least one dose of furosemide [43]. These data underscore the prevalent reliance on furosemide as a therapeutic option in the management of conditions affecting preterm infants in NICUs. Based on previous studies, primarily animal models, it was demonstrated that the impact of furosemide on the production of PGE2 in the cortical portion of the thick ascending limb of the loop of Henle is dose-dependent [19]. Given that PGE2 has been shown to delay the closure of DA [20,21], it can be hypothesized that in VLBW preterm infants, a higher cumulative dose of furosemide could lead to increased renal production of PGE2, thereby extending the time to spontaneously close DA. However, quantifying serum PGE2 levels in our setting is challenging due to the absence of the necessary assay equipment in our hospital. Upon reviewing the literature, we found no study directly investigating the relationship between furosemide dosage and spontaneous ductal closure in preterm infants, highlighting a gap in the current understanding of this association. An RCT targeting infants who received indomethacin therapy before reaching 96 h of age, with a BBW of less than 2000 g and a GA of less than 34 weeks, found that furosemide did not influence the rate of PDA closure induced by indomethacin but did increase the incidence of acute kidney injury [44]. This study highlights the complexities of managing PDA in preterm infants, where furosemide, despite its widespread use for various indications, does not contribute to the effectiveness of indomethacin in closing the PDA but poses a risk for renal complications. Another retrospective study discussed the association between furosemide exposure and the risk of PDA treatment (either pharmacologically with indomethacin or ibuprofen, or through surgical intervention) among hospitalized VLBW infants, showing exposure to furosemide did not increase the need for PDA treatment [45]. Similarly, our study revealed that there was no significant link between the cumulative furosemide dose and the time that spontaneous ductal closure took. These findings suggest that, unlike what animal models have demonstrated, there might not be a causal relationship between furosemide exposure and the process of ductal closure in VLBW preterm infants.

Upon analyzing our data, we observed there was no difference in clinical outcomes between infants diagnosed with PDA and those without. However, subgroup analysis showed that infants with PDA who were exposed to furosemide had a higher risk of developing BPD and ROP compared to those with PDA but without furosemide exposure. Following both univariate and multivariate analyses focused on BPD and ROP, GA emerged as the most significant risk factor. This finding indicates that the more premature infants are, the greater vulnerability they have to develop BPD and ROP, primarily due to their underdeveloped organ systems. The pathophysiology of BPD involves damage, injury, and inflammation affecting the delicate and immature lung tissue. Other risk factors contributing to the development of BPD, in addition to lower GA, include intrauterine growth restriction, exposure to maternal smoking, the use of mechanical ventilation, oxygen toxicity, and postnatal infections [46]. These factors underscore the complex interplay of environmental and biological influences that exacerbate the risk for these conditions in preterm infants. The association between PDA and the development of BPD remains uncertain. Studies conducted prior to 2000 consistently suggested a link between PDA and BPD, but these findings have been called into question due to concerns about study design [47]. More recent investigations into the use of prophylactic indomethacin to prevent PDA have yielded conflicting results regarding its impact on the development of BPD. One study reported that prophylactic indomethacin significantly reduced the incidence of BPD, with a risk ratio of 0.68 and a 95% confidence interval of 0.46 to 0.89 [48]. In contrast, another study found no significant association between prophylactic indomethacin use and the development of BPD, with an odds ratio of 0.89 and a 95% confidence interval of 0.72 to 1.10 [49]. These divergent findings highlight the complexity of understanding the role of PDA in BPD development and the need for further research to clarify this relationship.

Regarding the risk of ROP, identified risk factors include low BBW, low GA, the use of mechanical ventilation for more than 7 days, a high volume of blood transfusions, and surfactant administration. Conversely, protective factors against ROP include NEC, maternal pre-eclampsia, prenatal administration of betamethasone, vitamin E supplementation, and phototherapy [50]. Our findings are partially inconsistent with previous research, which could be attributed to our relatively smaller sample size. Additionally, our study suggests that PDA does not significantly contribute to the development or progression of ROP.

The strength of our study is that it might be the first study to analyze the impact of furosemide usage on spontaneous closure of PDA. Clinically, our findings provide supportive data for the conservative management of PDA, presenting a valuable perspective for healthcare providers. However, the retrospective design of our study introduces certain limitations. We acknowledge the limitation due to the small sample size of infants with PDA who did not receive furosemide. Our data collection spanned only the past three years from 2019 to 2022, leading to a relatively smaller sample size. This was intentionally chosen to minimize the variability in care quality that could potentially affect the data. We recognize the need for caution in drawing conclusions from this sample. Furthermore, quantifying fluid status was challenging due to different practices among attending neonatologists regarding daily fluid restrictions for infants. It was also difficult for us to express the furosemide dose of each infant as milligrams per kilogram because it is hard to decide the most suitable body weight for standardization, so we chose to use the cumulative dose of furosemide. Although criteria for hsPDA were established, the timing for initiating furosemide treatment varied based on individual clinicians’ judgments. Despite the limitations, our findings offer valuable insights into the relationship between furosemide exposure and ductal closure in the natural progression of VLBW preterm infants. To build upon this initial understanding, further research should aim for a larger sample size, employ more objective criteria for hsPDA, and achieve more accurate quantification of fluid status to further elucidate the role of furosemide in the management of PDA.

## 5. Conclusions

The prescription of furosemide in VLBW preterm infants does not influence the likelihood or PMA of spontaneous ductal closure. Additionally, the cumulative dose of furosemide does not significantly impact the natural progression of ductal closure. It is understandable that the duration of furosemide exposure correlated with the duration of ductal patency, as furosemide is prescribed until the closure of the DA when an infant exhibits symptoms or signs of hsPDA. The most significant factor influencing the time for spontaneous ductal closure is likely GA, which demonstrates a negative association.

## Figures and Tables

**Figure 1 children-11-00610-f001:**
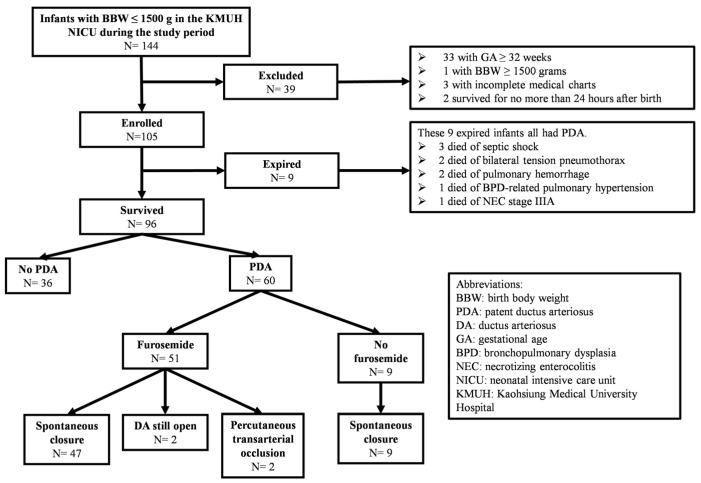
Flow chart of the study infants during the study period.

**Table 1 children-11-00610-t001:** Characteristics of VLBW preterm infants with/without PDA.

Characteristics	PDA (N = 60)	No PDA (N = 36)	*p* Value
Birth body weight (gm) Mean (SD)	992.4 (267.3)	1174.5 (207.3)	0.001
Gestational age (weeks) Mean (SD)	27.1 (2.2)	28.4 (2.0)	0.003
1st min Apgar score Mean (SD)	4.0 (1.9)	4.8 (1.8)	0.025
5th min Apgar score Mean (SD)	5.9 (2.2)	6.7 (2.0)	0.098
Male, n (%)	32 (53.3)	20 (55.6)	0.832
SGA, n (%)	2 (3.3)	0 (0.0)	0.268
LGA, n (%)	2 (3.3)	2 (5.6)	0.598
Antenatal steroid exposure, n (%)	49 (81.7)	32 (88.9)	0.345
Intubation at birth, n (%)	34 (56.7)	10 (27.8)	0.006

SD: standard deviation; VLBW: very low birth weight; PDA: patent ductus arteriosus; SGA: small for gestational age; LGA: large for gestational age.

**Table 2 children-11-00610-t002:** Characteristics of VLBW preterm infants with PDA with/without furosemide exposure.

Characteristics	Furosemide (N = 51)	No Furosemide (N = 9)	*p* Value
Birth body weight (gm) Mean (SD)	931.7 (238.8)	1336.4 (117.5)	<0.001
Gestational age (weeks) Mean (SD)	26.6 (2.0)	29.6 (1.2)	<0.001
1st min Apgar score Mean (SD)	3.8 (1.9)	5.0 (1.5)	0.069
5th min Apgar score Mean (SD)	5.7 (2.3)	7.2 (1.0)	0.002
Male, n (%)	27 (52.9)	5 (55.6)	0.885
SGA, n (%)	2 (3.9)	0 (0.0)	0.546
LGA, n (%)	2 (3.9)	0 (0.0)	0.546
Antenatal steroid exposure, n (%)	42 (82.4)	7 (77.8)	0.744
Intubation at birth, n (%)	31 (60.8)	3 (33.3)	0.125
PDA spontaneous closure, n (%)	47 (92.2)	9 (100.0)	0.384
Time taken for PDA spontaneous closure * (days) Mean (SD)	55.1 (33.3)	28.0 (18.4)	0.022
PMA at PDA spontaneous closure * (weeks) Mean (SD)	35.0 (4.1)	34.1 (3.5)	0.558

* The furosemide group excluded 4 infants who did not have PDA spontaneous closure, so the case number was 47. SD: standard deviation; VLBW: very low birth weight; PDA: patent ductus arteriosus; SGA: small for gestational age; LGA: large for gestational age; PMA: postmenstrual age.

**Table 3 children-11-00610-t003:** Regression analysis of factors associated with the time taken for spontaneous ductal closure in VLBW preterm infants with furosemide exposure.

Factors	Univariate						Multivariate					
	RC	Beta Value	R, R^2^	95% CI		*p* Value	RC	Beta Value	R, R^2^	95% CI		*p* Value
				Lower	Upper					Lower	Upper	
Furosemide cumulative dose	0.458	0.199	0.458, 0.209	0.083	0.315	0.001	−0.090	−0.039	0.648, 0.420	−0.222	0.144	0.668
Duration of furosemide treatment	0.603	0.741	0.603, 0.364	0.447	1.035	<0.001	0.547	0.672		0.083	1.261	0.026
Birth body weight	−0.365	−0.050	0.365, 0.133	−0.089	−0.012	0.012	0.193	0.027		−0.026	0.079	0.317
Gestational age	−0.525	−8.725	0.525, 0.276	−12.968	−4.483	<0.001	−0.384	−6.370		−13.066	0.326	0.062
1st min Apgar score	−0.167	−2.902	0.167, 0.028	−8.044	2.240	0.262						
5th min Apgar score	−0.125	−1.796	0.125, 0.016	−6.075	2.482	0.402						
Male	0.105	6.937	0.105, 0.011	−12.734	26.607	0.481						
Antenatal steroid exposure	0.116	10.170	0.116, 0.013	−15.963	36.302	0.437						
Intubation at birth	0.152	10.278	0.152, 0.023	−9.828	30.383	0.309						

RC: regression coefficient; CI: confidence interval; VLBW: very low birth weight.

**Table 4 children-11-00610-t004:** Clinical outcomes of VLBW preterm infants.

Outcomes	PDA				No PDA (N = 36)	*p* Value ^#^
	Furosemide (N = 51)	No Furosemide (N = 9)	*p* Value *	Total (N = 60)		
IVH, n (%)	31 (60.8)	5 (55.6)	0.768	36 (60.0)	24 (66.7)	0.514
BPD, n (%)	40 (78.4)	4 (44.4)	0.034	44 (73.3)	21 (58.3)	0.128
NEC, n (%)	1 (2.0)	1 (11.1)	0.159	2 (3.3)	1 (2.8)	0.880
ROP, n (%)	39 (76.5)	3 (33.3)	0.009	42 (70.0)	20 (55.6)	0.152
PVL, n (%)	1 (2.0)	1 (11.1)	0.159	2 (3.3)	3 (8.3)	0.286
Pulmonary air leak syndrome, n (%)	2 (3.9)	0 (0.0)	0.546	2 (3.3)	0 (0.0)	0.268
Post-hemorrhagic hydrocephalus, n (%)	1 (2.0)	0 (0.0)	0.672	1 (1.7)	2 (5.6)	0.289

* This *p* value is for the groups of furosemide and no furosemide. ^#^ This *p* value is for the groups of PDA and no PDA. VLBW: very low birth weight; PDA: patent ductus arteriosus; IVH: intraventricular hemorrhage; BPD: bronchopulmonary dysplasia; NEC: necrotizing enterocolitis; PVL: periventricular leukomalacia; ROP: retinopathy of prematurity.

**Table 5 children-11-00610-t005:** Regression analysis of factors associated with BPD in VLBW preterm infants.

Factors	Univariate						Multivariate					
	RC	Beta Value	R, R^2^	95% CI		*p* Value	RC	Beta Value	R, R^2^	95% CI		*p* Value
				Lower	Upper					Lower	Upper	
Birth body weight	−0.402	−0.001	0.402, 0.161	−0.001	0.000	<0.001	0.081	0.000	0.539, 0.291	0.000	0.001	0.608
Gestational age	−0.486	−0.104	0.486, 0.237	−0.143	−0.066	<0.001	−0.403	−0.087		−0.154	−0.020	0.012
1st minute Apgar score	−0.313	−0.078	0.313, 0.098	−0.127	−0.030	0.002	−0.100	−0.025		−0.108	0.058	0.550
5th minute Apgar score	−0.311	−0.068	0.311, 0.097	−0.111	−0.025	0.002	−0.105	−0.023		−0.095	0.049	0.529
Furosemide exposure	0.349	0.437	0.349, 0.122	0.197	0.678	<0.001	0.182	0.228		−0.024	0.480	0.076
PDA	0.155	0.150	0.155, 0.024	−0.045	0.345	0.131						
Male	0.125	0.117	0.125, 0.016	−0.074	0.308	0.226						
SGA	0.101	0.330	0.101, 0.010	−0.337	0.997	0.329						
LGA	−0.079	−0.185	0.079, 0.006	−0.662	0.293	0.444						
Antenatal steroid exposure	−0.052	−0.067	0.052, 0.003	−0.330	0.197	0.616						
Intubation at birth	0.143	0.135	0.143, 0.021	−0.056	0.325	0.163						

VLBW: very low birth weight; BPD: bronchopulmonary dysplasia; RC: regression coefficient; CI: confidence interval; PDA: patent ductus arteriosus; SGA: small for gestational age; LGA: large for gestational age.

**Table 6 children-11-00610-t006:** Regression analysis of factors associated with ROP in VLBW preterm infants.

Factors	Univariate						Multivariate					
	RC	Beta Value	R, R^2^	95% CI		*p* Value	RC	Beta Value	R, R^2^	95% CI		*p* Value
				Lower	Upper					Lower	Upper	
Birth body weight	−0.524	−0.001	0.524, 0.275	−0.001	−0.001	<0.001	−0.115	0.000	0.602, 0.363	−0.001	0.000	0.429
Gestational age	−0.599	−0.131	0.599, 0.358	−0.167	−0.095	<0.001	−0.518	−0.114		−0.177	−0.050	0.001
Furosemide exposure	0.253	0.325	0.253, 0.064	0.071	0.579	0.013	−0.030	−0.038		−0.278	0.201	0.751
1st minute Apgar score	−0.180	−0.046	0.180, 0.032	−0.098	0.006	0.079						
5th minute Apgar score	−0.115	−0.026	0.115, 0.013	−0.071	0.020	0.266						
PDA	0.146	0.144	0.146, 0.021	−0.056	0.345	0.155						
Male	−0.069	−0.066	0.069, 0.005	−0.263	0.130	0.503						
SGA	−0.044	−0.149	0.044, 0.002	−0.834	0.536	0.667						
LGA	0.045	0.109	0.045, 0.002	−0.381	0.598	0.660						
Antenatal steroid exposure	−0.079	−0.104	0.079, 0.006	−0.373	0.165	0.446						
Intubation at birth	0.157	0.150	0.157, 0.025	−0.044	0.344	0.127						

VLBW: very low birth weight; ROP: retinopathy of prematurity; RC: regression coefficient; CI: confidence interval; PDA: patent ductus arteriosus; SGA: small for gestational age; LGA: large for gestational age.

## Data Availability

The datasets generated and/or analyzed during the present study are available from the corresponding author on request. The data are not publicly available due to privacy.

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
