# Peer review of "Furosemide and Ductus Arteriosus Closure in Very-Low-Birth-Weight Preterm Infants: A Comprehensive Retrospective Study"

_children, 2024, doi:10.3390/children11050610_

Round 1

Reviewer 1 Report

Comments and Suggestions for Authors

Title: consider revising the title

Abstract:

1.       Typo in line 18

2.       Line 20 – perhaps state the coefficient value rather than just the p value

3.       Consider revising line 21.

4.       Consider stating the cumulative dose of furosemide

Methods:

1.       Are there any echocardiographic parameters that the team uses to define haemodynamic significance?

2.       Is furosemide commenced based on clinical grounds or is there a baseline echocardiogram? Please comment.

3.       What dose of furosemide is used?

4.       What is the indication for an echocardiogram in your unit? It would be helpful to describe what is your routine practice in your unit for the reader to understand.

5.       What is the rationale for using furosemide to manage the PDA. Is furosemide used in your unit for improving the respiratory status or is it purely for managing the PDA?

Results:

1.       Time of PDA closure was higher in furosemide group when compared to group that received no furosemide. Could this be explained by furosemide administration?

2.       Why is that the total duration of furosemide was found to be significant predictor for PDA closure in multivariate regression but not cumulative dose of furosemide?

3.       Please provide the cumulative doses of furosemide for both the groups.

Discussion:

1.       The authors have clearly demonstrated through multivariate regression analysis that the duration of furosemide treatment was associated with spontaneous ductal closure in VLBW infants. Therefore, the first line of the discussion seems to contradict the findings in the study. Please comment.

2.       Line 206 – this is very difficult to comprehend with the manuscript in it current form. The authors have not described what the indications are for echocardiograms in their unit. Please describe what is standard practice including indications for echocardiogram in this group of infants.

3.       The sample size of infants with PDA who do not receive furosemide is very small. So caution should be exercised in drawing out too much conclusion using this sample.

4.        

References:

1.       Reference number 24 link does not work!

Comments on the Quality of English Language

Well written with some typos.

Reviewer 2 Report

Comments and Suggestions for Authors

This study, which primarily examines the effect of Furosemide in the treatment of PDA, is an explanatory experimental research. Although the results of this study actually reveal the known problems with PDA, it aimed to try a different treatment method.I have a few correction suggestions for the work.

1.First, replace P in the entire text with p.

2.You should write more details about sample calculation and sample selection. You can use g power in the sample calculation. You can do this by taking standard deviations from similar studies

3.Beta values are important in regression analyses. At the same time, r and r squared are given separately. I recommend adding these to your tables.

4.Why don't you mention the different drugs used for PDA in the introduction. You can talk about the experiments performed. In this way, you can support that it is not wrong to try different medications.

5. Did you leave premature babies to their own fate for whom you did not apply furosemide? Or did they receive a classical prostaglandin treatment? There is nothing clear about this in the text. Please clarify this issue. It's a serious situation. it is not ethical.

6.In the conclusion section, you mentioned that you found furusemide not effective. Why don't you mention that this is an important finding and that new interventions are needed for this drug? As a result, there is a slight relationship. Perhaps if furosemide and prostaglandin are given together, a faster closure may occur or different alternatives can be tried. New results can be obtained through animal experiments.

Round 2

Reviewer 1 Report

Comments and Suggestions for Authors

I am happy with the authors responses.

Reviewer 2 Report

Comments and Suggestions for Authors

The author made as many corrections as he could. In this form, this work can be published